# Synergy of higher education resources and digital infrastructure construction in China: Regional differences, dynamic evolution and trend forecasting

Ying Xie[1], Minglong Zhang[2]*

**1** Student Affairs, Chongqing Business Vocational College, Chongqing, China, **2** School of Economics and Business Administration, Chongqing University of Education, Chongqing, China

* zhangml@cque.edu.cn

## Abstract

The deep integration of higher education with digital technology represents an inevitable trend, and evaluating the interplay between higher education resources (HER) and digital infrastructure construction (DIC) holds significant value for advancing the development of digital higher education and mitigating regional disparities in China. This study establishes two comprehensive evaluation frameworks for HER and DIC. Panel data from 31 provinces, spanning the period from 2011 to 2020, are utilized for analysis. The coupling coordination degree (CCD) model is employed in this work to evaluate the synergy between HER and DIC in China. Furthermore, we analyze the regional differences, spatial distribution, and trend evolution of this synergy. The study results revealed that there is an initial decrease followed by an increase in the synergy between HER and DIC, and the overall CCD is at a moderate coordination, with the mean CCD of the eastern region being significantly higher than that of the other three regions, and the inter-regional difference is the main source of regional disparity in this synergy. The current state of synergistic development reveals a slight inclination towards multi-polarization, although the disparity in regional development was decreasing. Additionally, there is an observed convergence in the coordinated development of HER and DIC, with spatial factors playing a significant role. These findings offer empirical support for efforts to enhance the integration of HER and DIC, reduce regional disparities in higher education, and foster sustainable development in China's higher education sector.

## 1. Introduction

The COVID-19 pandemic in early 2020 significantly impacted higher education [1]. Several international organizations, including the United Nations Educational, Scientific and Cultural Organization (UNESCO), the United Nations (UN), the Organization for Economic Co-operation and Development (OECD), and the World Bank, proposed recommendations advocating for the advancement of information and communication technology (ICT) and virtual

**Funding:** The author(s) received no specific funding for this work.

educational models [2–5], and promoting digital environments in higher education [6], to mitigate the negative impacts of the pandemic. Meanwhile, The International Association of Universities (IAU) had recently released a publication entitled "*Transforming Higher Education in a Digital World for the Global Common Good*". This publication advocates for a concentrated effort in digitizing higher education, with a specific emphasis on ethical considerations, inclusivity, and the pursuit of initiatives that prioritize the welfare of the global community. It is essential that these goals are achieved through the training provided [7]. EDUCAUSE published *Top 10 IT Issues*, *2020*: *The Drive to Digital Transformation Begins*, describing the key issues driving digital transformation in higher education [8]. In August 2021, the Ministry of Education in China gave its approval for Shanghai to serve as a pilot zone for the transformation of digital education [9]. Subsequently, on January 16–17, 2022, a national education conference was held to implement the initiative for digital education strategy [10]. This series of related policies has catalyzed the demand for digital construction in higher education, accelerated the integration of higher education and digital technology, and thus has become a new hotspot that has attracted much attention in the field of higher education, which will surely become a major breakthrough in the reform and development of higher education in China.

Nowadays, the use of digital technology in universities is gradually transforming the traditional methods and habits of teaching and learning. Evidence suggests that combining educational resources with more advanced technology will promote regional productivity growth [11]. Advanced technologies such as 5G, artificial intelligence, big data, Internet of Things, and Metaverse are providing a broader platform for sharing knowledge, teaching methods, and educational resources. This is contributing to a comprehensive digital transformation in the field of higher education and may give rise to a new model of higher education in China. Therefore, it is essential to adapt to the changing landscape of education and teaching in the age of information technology, promote the seamless integration of ICT with teaching, and facilitate the high-quality development of higher education. This is a key challenge that needs to be urgently addressed and resolved.

Previous studies have found that there is a relatively obvious imbalance higher education development in China [12–14]. Meanwhile, considering the economic development in different regions, researchers have also found significant regional differences in digital development [15, 16]. This may bring about variability in the synergistic development of higher education and digitization in different regions. Therefore, this work examines the coupling and coordination degree (CCD) of higher education resources (HER) and digital infrastructure construction (DIC). The combined effect of this synergy has the potential to invigorate the fusion and execution of digital technology within the realm of higher education, while also delving into its forthcoming prospects. Meanwhile, the digital infrastructure construction supports the integration of higher education resources by overcoming limitations in spatial and temporal conditions. This has a positive impact on the joint integration of the two and reduces regional developmental differences.

The remaining sections of this paper are structured in the following manner. Section 2 provides a comprehensive literature review; Section 3 states the synergistic mechanisms of HER and DIC; Section 4 describes the methodology; Section 5 lists the results and empirical analysis; and Section 6 draws the conclusions and countermeasures.

## 2. Literature review

An increasing number of universities have been developing "digital education" with the help of ICT [17]. A richer exploration of the link between higher education and digitization has been undertaken by scholars, including the following aspects:

Firstly, the need for digital technologies in higher education. A multitude of academic investigations have confirmed that the COVID-19 pandemic has compelled universities to embrace online platforms to maintain their educational systems, highlighting the need for digital integration in global education [18, 19]. Penprase [20] argued that using advanced tech like projectors and computers in higher education can enhance student engagement and foster active participation by providing an immersive learning experience. Ozdamli and Cavus [21] highlighted the importance of feedback loops in digital classrooms, noting that they allow for real-time instructor feedback, which is crucial for student learning.

Secondly, the positive impact of digital technology in higher education. Regarding the utilization of teaching resources, Akbar [22] asserted that educators could track the use of resources and student activities through digital technology, and that the benefits of archiving and preserving teaching resources could help in the development of teaching resources. The advent of digital technology has led to noteworthy advancements in the field of education, manifesting in enhanced learning methodologies and greater accessibility to educational materials [23]. As posited by Henderson et al. [24], the incorporation of digital technology has resulted in enhanced accessibility to higher education through a transformative redefinition of the roles traditionally assumed by educators and learners. Pedro et al. [25] argued that the widespread use of digital technologies can eliminate the need for physical space and change how educational content is transmitted between educators and learners, both at a pedagogical and organizational level. For example, physical laboratories may be replaced by simulation-based laboratories [26] or virtual and augmented reality [27]. Marks and Thomas [28] argued that virtual reality technology could help to create online libraries that facilitate interaction between faculty, students and researchers. In terms of the effectiveness of teaching, Martin et al. [29] found that the use of learning management systems was considered highly important. Additionally, the study showed that teachers can foster curiosity in students through the use of digital technologies such as e-Learning and ICT, thereby enhancing the learning process [30, 31]. From the students' perspective, scholars have argued that digital technologies have not only impacted their social lives but have also facilitated collaboration in the learning process and promoted independent learning [32–34]. Marinagi et al. [35] stated that students benefited from new learning opportunities and additional support in classrooms equipped with various computing devices and technologies.

Thirdly, the challenges of digital transformation in higher education. UNESCO released *Reimagining our Future Together: a New Social Contract for Education*, stating that digital technologies have enormous transformative potential, but the process of integrating teaching and learning in universities and colleges with digital technologies still presents many challenges [36]. For example, the digital divide can widen the problem of uneven development in higher education. Many students in remote areas lack access to distance online learning and must rely on limited technological resources. This highlights the widespread technological and educational inequalities on a global scale [37]. According to Du Toit and Verhoef [38], disadvantaged groups find it challenging to effectively use digital technology, emphasizing the need to address the issue of the digital divide in higher education. Tulinayo et al. [39] explored the use of digital technology in resource-constrained colleges and universities and discovered that limited access to this technology in these institutions contributes to the low level of its use and acceptance among students in their learning process. Additionally, the digital teaching and learning competences of teachers and students are also a great challenge. Akbar [22] argued that digital technology had clear advantages in teaching and learning, but teachers need to invest a lot of time and effort in mastering it at the beginning of adopting this technology for teaching and learning. In online courses in higher education, "skipping" or missing classes becomes easier for students who lack digital learning self-discipline [40]. Finally, there is a lack

of direct interaction between instructors and students and online courses do not satisfy students' acquisition of practical knowledge. Njoku [41] claimed that practical knowledge acquisition by university students in online courses is hindered by the absence of digital pedagogical competence of some instructors, particularly when the number of practical courses surpasses theoretical ones. Moreover, several studies have shown that online teaching is not suitable for teaching practical skills in the absence of seminars or face-to-face interactions [19, 42].

In essence, researchers have extensively studied the importance, beneficial effects, and difficulties associated with incorporating digital technologies in higher education. However, there is limited scholarly attention given to the simultaneous advancement of higher education resources (HER) and the establishment of digital infrastructure (DIC), as well as regional variations. Yet, the integration of higher education and digitization is actually the crucial aspect of digital transformation in higher education. This paper develops a comprehensive indicator system for HER and DIC, based on an explanation of the intertwined and coordinated relationship between digitalization and higher education. Using panel data from 31 provinces in mainland of China from 2011 to 2020, this work employs the coupling coordination degree (CCD) model to assess the synergy of HER and DIC. Additionally, the study also analyzes regional differences, distribution dynamics, and long-term evolution trends in this synergistic development by various analytical tools such as Dagum's Gini coefficient, kernel density estimation, and Markov chain analysis. The main findings of this study are: (1) The synergy between HER and DIC is at the moderate coordination and varies significantly across regions. (2) The regional differences in this synergy are mainly due to inter-regional difference. (3) There is an observed pattern of initially declining and subsequently rising synergy, along with a minor inclination towards multi-polarization. However, the regional disparities in development are diminishing. (4) This synergy is distinguished by the phenomenon of club convergence, wherein spatial factors play a significant role in influencing it.

## 3. Coupling coordination mechanism of HER and DIC

The synergy between HER and DIC denotes their interaction and influence on each other. It has serious imbalances among regions due to the heterogeneity of higher education development and digitization as a result of the variability of economic development and geographic constraints in each region. A robust digital infrastructure can facilitate digital technology has a strong penetration effect, weaken the physical access limitations, break the barrier of uneven distribution of higher education resources in time and space, with more diversified connections to transform the previous teaching mode, which aims to realize the sustainable development goals in higher education. Thus, a profound correlation exists between higher education and digitalization, which establishes the basis for the synergy of higher education resources and digital infrastructure.

New digital technologies are one of the driving forces behind the modernization of Chinese higher education in three main ways:

Firstly, the fusion of digitization and higher education has given rise to innovative frameworks, enriching the digital proficiency of both instructors and students in the realm of higher education. The expeditious progression of digital infrastructure has elevated the degree of informatization within the realm of higher education, consequently enhancing the overall digital milieu. The utilization of new information technology in higher education contributes to the enhancement of digital literacy among teachers and students. As digital technology continues to advance, the use of digital tools for teaching and digital resources for learning will significantly enhance the digital literacy of teachers and students, laying a strong foundation for the seamless integration of information technology and higher education.

Secondly, the advancement of digital infrastructure will mitigate the regional disparity in distributing resources for higher education, therefore promoting equal opportunities and augmenting the quality. Due to the vastness of China, limited by the regional economic level and geographical conditions, quality higher education resources have obvious scarcity, which leads to obvious differences between regions. The advancements in digital technology have the potential to greatly bridge the gap between different locations and time periods in higher education. This can help address the issue of unequal distribution of educational resources caused by factors such as geography and time. Through the use of high-quality online courses, micro-courses, virtual simulation experiments, and other digital teaching and research methods, we can promote the sharing of educational resources and facilitate the effective integration of these resources. Ultimately, this will contribute to the digital transformation in universities and colleges.

Thirdly, digital infrastructure can aid the facilitation of smart campuses in universities, thereby enhancing the adaptability and competitiveness of higher education. The establishment of intelligent campuses in higher education institutions assumes a critical role in enriching the caliber of pedagogy, scholarly exploration, and comprehensive administration of higher education [43]. Moreover, it constitutes a fundamental component of the advancing terrain of teaching and research in the era of digitalization. Therefore, it will rely on the support of digital infrastructure based on artificial intelligence, big data analytics and cloud computing. In addition, the construction of digital infrastructure has brought about innovations in the internal efficiency of higher education, reduced constraints such as information asymmetry and information lag, strengthened the adaptability of higher education, and enhanced its competitiveness and attractiveness in the world's higher education system.

While the application of digital technologies helps to integrate higher education resources, higher education also provides power for digital infrastructure construction in the field of education.

First of all, higher education has a significant role to play in promoting the installation of digital infrastructure via the implementation of technology. Digital infrastructure refers to the creation and implementation of modern information technology systems, which require a suitable platform for their application. Higher education institutions provide such a platform for integrating digital technology into the education sector. These institutions have enhanced the alignment of their academic disciplines and programs with the demands of digital transformation. They achieve this by offering and optimizing courses related to the construction of digital infrastructure, such as computer science and technology, integrated circuit design and integrated systems, data science and big data technology, as well as Internet of Things engineering and artificial intelligence.

Moreover, higher education institutions play a crucial role in nurturing skilled individuals for digital advancements and enhancing the level of digital technology. By establishing and improving mechanisms to motivate talent, higher education aims to attract and assemble proficient individuals and teams in the field of digitalization. This, in turn, facilitates the cultivation of skilled educators and the creation of efficient digital teaching platforms, thereby allowing for successful implementation of educational resources and digital technology in higher education institutions. Consequently, there is an alignment between talent training and research and development efforts in digital technology, resulting in the cultivation of more human resources essential for digital transformation. Ultimately, this forms the fundamental basis for the sustainable development of digitalization.

Finally, in realizing digital cooperation between industry, universities and research, university resources play an important role. Colleges and universities utilize the construction of digital industrial parks to effectively amalgamate higher education resources, incubate digital

industries, expedite the application of digital technology, encourage industry-university-research collaboration on digitalization, and gradually form a cluster effect of higher education with the digital industry. Such initiatives accelerate the profound amalgamation of higher education with modern information technology.

## 4. Methodology

### 4.1 Evaluation indicators system

**4.1.1 Higher education resources.** Educational resources are the essential elements required to facilitate educational activities, also referred to as "educational economic conditions". These conditions encompass the human, material, and financial resources that are utilized and consumed during the educational process. Higher education is a crucial component of the education system, playing a significant role in China's education reform and development. In this study, we have constructed an evaluation indicator system for China's higher education resources, based on the explanatory power of indicators and the availability of data [43, 44]. This system is organized into three dimensions: faculty allocation, funding input, and material resources, as presented in Table 1.

**4.1.2 Digital infrastructure construction.** Currently, the relevant data of DIC could not be obtained directly from the regional statistical databases. Some researchers have adopted a single index to characterize DIC, such as Internet penetration [45]. However, a single variable can only vaguely reflect the status of DIC, and does not represent its actual level in multiple dimensions. Considering that DIC is mainly manifested by the sharing of new-generation information technology and networks. By referring to previous research [46, 47] and adhering to the principles of objectivity, comprehensibility and accessibility, this work identifies eight indicators to assess DIC. These indicators include the number of mobile phone base stations, mobile phone switch capacity, length of long-distance optical cable line, mobile phone penetration rate, number of Internet domain names, number of Internet pages, number of Internet

**Table 1. Evaluation indicator system of higher education resources.**

| 1st-Class indicator | 2nd-Class indicator | Evaluating indicator | Indicator properties |
|---|---|---|---|
| Faculty allocation | Student and teacher ratio | | - |
| | Proportion of teachers with associate professor or above (%) | Number of teachers with associate professor or above / Number of full-time teachers | + |
| | Proportion of faculty with doctoral degrees (%) | Number of doctoral teachers / Number of full-time teachers | + |
| Funding input | Per capita education expenditure per student (yuan/person) | Expenditure on education / Number of students in universities and colleges | + |
| | Proportion of expenditure on education in the fiscal budgetary (%) | Public budget expenditure on higher education / Local fiscal general budget expenditure | + |
| | Per capita R&D expenditure in colleges and universities (10 yuan/person year) | Intramural expenditure on R&D in higher education / Full-time equivalent of R&D personnel in higher education | + |
| | Fixed assets per student in colleges and universities (yuan/person) | Fixed assets of colleges and universities / Number of students in universities and colleges | + |
| Material resources | Area of higher education campus per student (m²/person) | Area of higher education premises / Number of students in universities and colleges | + |
| | Number of books per student (volume/person) | Number of books / Number of students in universities and colleges | + |
| | Number of teaching computers per 100 students (set/100 persons) | Number of teaching computers / number of students in universities and colleges | + |

Note: "+" and "-" denote positive and negative indicators.

broadband access ports, and Internet penetration rate. To evaluate the level of DIC, we apply the entropy method to each region's data.

## 4.2 Methods

**4.2.1 Coupling coordination degree (CCD) model.** To evaluate the integration of HER and DIC, this paper employs CCD model to calculate their synergistic development. Among them, the coupling model of the two subsystems interacting with each other is Eq (1):

$$C = \frac{2\sqrt{U_{HER} \times U_{DIC}}}{U_{HER} + U_{DIC}} \tag{1}$$

Where C represents the degree of coupling, and $U_{HER}$ and $U_{DIC}$ are two subsystems responsible for higher education resources and digital infrastructure construction, respectively. The specific measurements are the following:

The first step is to standardize the raw data using the method of extreme variance:

$$x'_{ijk} = \frac{x_{ijk} - x_{\min}}{x_{\max} - x_{\min}} \tag{2}$$

Where $x_{ijk}$ denotes the $k$th indicator for region $j$ in year $i$, $x_{\max}$ and $x_{\min}$ represent the indicator's upper and lower limits, respectively.

The second step is to calculate the value of entropy for each indicator:

$$y_{ijk} = x'_{ijk} / \sum_i \sum_j x'_{ijk} \tag{3}$$

$$e_k = -(1/\ln(tm)) \times \sum_i \sum_j y_{ijk} \ln(y_{ijk}) \tag{4}$$

Where $y_{ijk}$ denotes the proportion of indicators, $e_k$ represents the information entropy, $t$ and $m$ stands for the number of years and regions, respectively.

The third step is to measure the weights of the indicators:

$$g_k = 1 - e_k \tag{5}$$

$$\omega_k = g_k / \sum_{i=1}^n g_k \tag{6}$$

Where $g_k$ is the entropy redundancy of the indicator and $\omega_k$ is the weight of the indicator.

Finally, we can obtain the composite index of the subsystem. Based on the weights of each index mentioned above, the composite evaluation values of HER and DIC are derived respectively:

$$U = \sum_{k=1}^n \omega_k x'_{ijk} \tag{7}$$

Since the coupling model just only evaluates the degree of interaction between systems, it cannot measure their coordination. We therefore present the coupled coordination model, which is formulated:

$$D = \sqrt{C \times T} \quad T = \alpha U_{HER} + \beta U_{DIC} \tag{8}$$

Where $T$ is the degree of coordination, $\alpha$ and $\beta$ are the weight coefficients of higher education

**Table 2. Classifications of coupling coordination levels.**

| Value Range | Type |
|---|---|
| D∈[0.00, 0.20) | Severe dissonance |
| D∈[0.20, 0.40) | Primary coordination |
| D∈[0.40, 0.60) | Moderate coordination |
| D∈[0.60, 0.80) | Good coordination |
| D∈[0.80, 1.00) | High quality coordination |

resources and digital infrastructure construction, respectively. We assign the same weight to the two subsystems, so $\alpha = \beta = 0.5$. The coupling coordination degree, denoted by $D$, ranges from 0 to 1, with a higher value indicating a stronger coupling coordination between HER and DIC. Table 2 presents the classification standard for $D$.

**4.2.2 Dagum's gini coefficient and its decomposition.** This method has been widely applied in the studies of spatial disequilibrium phenomena since it was proposed [48]. It provides a thorough evaluation of how subsamples are distributed, effectively dealing with the problem of overlap between sample data and regional sources of variation. It also surpasses the boundaries of traditional Gini coefficients and Theil index. As a result, this study employs this method to examine the regional disparities in the synergy of HER and DIC. $G$ represents the Dagum's Gini coefficient, which has the following definition:

$$G = \sum_{j=1}^{k} \sum_{h=1}^{k} \sum_{i=1}^{n_j} \sum_{r=1}^{n_h} |y_{ji} - y_{hr}| / 2n^2 \bullet \bar{y} \tag{9}$$

$$G_{jj} = \sum_{i=1}^{n_j} \sum_{r=1}^{n_j} |y_{ji} - y_{jr}| / 2\bar{y}_j \bullet n_j^2 \tag{10}$$

$$G_w = \sum_{j=1}^{k} G_{jj} p_j s_j \tag{11}$$

$$G_{jh} = \sum_{i=1}^{n_j} \sum_{r=1}^{n_h} |y_{ji} - y_{hr}| / n_j n_h (\bar{y}_j + \bar{y}_h) \tag{12}$$

$$G_{nb} = \sum_{j=2}^{k} \sum_{h=1}^{j-1} G_{jh} (p_j s_h + p_h s_j) D_{jh} \tag{13}$$

$$G_t = \sum_{j=2}^{k} \sum_{h=1}^{j-1} G_{jh} (p_j s_h + p_h s_j)(1 - D_{jh}) \tag{14}$$

Where $G_w$ is the contribution of intra-regional variation, $G_{nb}$ is the contribution of inter-regional variation, $G_t$ is the contribution of super variable density, and the sum of the three is the overall Gini coefficient $G$; $G_{jj}$ is the gap within region $j$, and $G_{jh}$ is the Gini coefficient between region $j$ and region $h$. $y_{ji}(y_{hr})$ indicates the synergy of HER and DIC in any of the regions within region $j(h)$. $\bar{y}$ is the overall average; n and k are the number of provinces and regions, respectively; $n_j(n_h)$ is the number of provinces within the $j(h)$the region, and $i$ and $r$

are different provinces within the region $j(h)$. $p_j = n_j/n$, $s_j = n_j \bar{y}_j / n\bar{y}_j$, $j = 1, 2, \cdots, k$. $D_{jh} = (d_{jh} - p_{jh})/(d_{jh} + p_{jh})$ represents the relative impact of the coupling coordination degree between two regions, indicating the mathematical expectation of the sum of all sample values in the region $j$ and $h$ satisfying condition $y_{ji} - y_{hr} > 0$. $p_{jh} = \int_0^\infty dF_h(y) \int_0^y (y - x)dF_j(x)$ is the first-order moment of hypervariable that represents the weighted average of the sum of all sample values in the $j$, $h$ region with condition $y_{hr} - y_{ji} > 0$.

**4.2.3 Kernel density estimation.** It is a commonly used method to examine the unevenness of spatial distribution. We utilize this method to examine the location, dynamics, extensibility, and polarization trends of CCD between two subsystems at both the national level and within four major regions. Following Parzen's (1962) [49], we assume that f(x) represents the density function in the synergy of HER and DIC in China:

$$f(x) = \frac{1}{Nh} \sum_{i=1}^{N} K(\frac{X_i - x}{h}) \tag{15}$$

Where $K(\bullet)$ denotes the kernel density function, the number of observations, denoted as $N$, and the bandwidth, represented by $h$, is inversely related to the accuracy of the estimation. Specifically, as the bandwidth decreases, the accuracy of the estimation increases. $X_i$ represents independent and identically distributed observations, x is the mean value of the observations. The study applies the Gaussian kernel density function for estimation purposes, with the particular equation demonstrated in Eq (16):

$$K(x) = \frac{1}{\sqrt{2\pi}} exp(-\frac{x^2}{2}) \tag{16}$$

**4.2.4 Markov chain.** The conventional Markov chain method examines how the synergy of HER and DIC changes in various regions. This is done by converting continuous data into discrete categories and analyzing the probability distribution and evolutionary pattern of each category. The fundamental model is defined as follows:

$$P\{X(t) = j | X(t-1) = i, X(t-2) = i_{t-2}, \cdots, X(0) = i_0\} = \{X(t) = j | X(t-1) = 1\} \tag{17}$$

$$P_{ij} = \frac{n_{ij}}{n_j} \tag{18}$$

Eq (17) describes the behavior of the 1st-order martensitic chain, where the likelihood of state $j$ occurring in random variable $X$ at time $t$ depends solely on the state of $X$ at time $t$-1. Eq (18) introduces $P_{ij}$ as the probability of the combined occurrence of HER and DIC in the specific region, transitioning from state $i$ at time $t$ to state $j$ at time $t$+1. Additionally, $n_{ij}$ represents the number of provinces that transition from state $i$ in the initial period to state $j$ in the subsequent period, while $n_j$ indicates the total number of provinces in the initial period that are in state $j$.

A spatial Markov chain is an extension of the traditional Markov chain model, where the influence of neighboring areas on the local area is taken into account. It analyzes the probability of the transfer of synergy between neighboring areas and the local area. Measuring the coupling coordination of neighboring provinces requires the help of a spatial weight matrix ($W$), $W$ is $\sum_j w_{ij} y_j$, where $w_{ij}$ is the elements in $W$, $i = j = 1, \ldots, n$, and $y_j$ is the coupling

coordination degree of region $j$. When considering spatial factors, it can be denoted as $P_{ij|\lambda}^{t,t+d}$ that represents the probability of local synergy transferring from state $i$ to state $j$ after a certain period $d$, based on the period t and spatial lag type $\lambda$. In this case, change in the CCD of two subsystems in neighboring types is called an upward (or downward) transfer, and it is called positive (or negative) jump transfer for changes across neighboring types. For instance, transitioning from a low level to a low-medium level is called an upward shift, while a shift from a low level to a medium-high or high level is known as a positive jump shift. Additionally, we can ascertain whether neighboring regions exert an influence on the transfer process of local synergy by comparing the transfer matrix probability magnitudes of the conventional Markov chain and the spatial Markovian chain. If $P_{12}^{t,t+d} > P_{12|1}^{t,t+d}$, it suggests that low-level neighborhoods will inhibit upward transfer of synergistic development in low-level areas. For any region $i$ and $j$, if $P_{ij|\lambda}^{t,t+d} = P_{ij}^{t,t+d}(\lambda = 1, 2, 3, 4)$, it indicates that no level of neighboring regions will have an impact on the transfer of local synergistic development. In other words, there are no spatial spillovers of this synergistic development among regions.

### 4.3 Data sources

This study focuses on examining the 31 provinces in mainland China from 2011 to 2020 as the research object. It is important to note that Hong Kong, Macao, and Taiwan are not included in this study due to limited availability of data. The data used in this study are obtained from various sources, including the China Education Statistical Yearbook, China Education Expenditure Statistical Yearbook, China Science and Technology Statistical Yearbook, China Statistical Yearbook, the National Bureau of Statistics (NBS) database, the statistical yearbooks of provinces, and additional data from the Wind database. The NBS division methodology was used to categorize the 31 provinces into four regions: East, Central, West, and Northeast.

## 5. Results and analysis

### 5.1 Analysis of results for CCD

Based on the above measurement formula, we obtain the coupled coordination degree of HER and DIC in 31 provinces in mainland China from 2011 to 2020. Fig 1 depicts the synergy of the two in each province.

Specifically, the main regions with relatively high synergistic development of HER and DIC are Beijing (0.8934), Shanghai (0.6556), Zhejiang (0.6423), Guangdong (0.6289), and Jiangsu (0.6045), whose average CCD is higher than 0.60, and are at the good coordination. Among them, the average CCD of Beijing is the highest (0.8934), achieving high quality coordination. It can be seen that the provinces with a higher average value of synergy are located in the eastern part of the region. The regions with lower synergy are mainly Inner Mongolia (0.3953), Anhui (0.3832), Yunnan (0.3660), Xinjiang (0.3647), Jiangxi (0.3638), Qinghai (0.3581), Gansu (0.3457), Guangxi (0.3437), Guizhou (0.3424), Shanxi (0.3377), Ningxia (0.3315), Hainan (0.3236) and Tibet (0.3058), the average CCD of these regions is lower than 0.40, which is at the primary coordination. It can be seen that most of the provinces at this level of synergy belong to the western region. Among them, the average CCD of Tibet is the lowest, which is only 0.3058. Additionally, the other provinces are at the moderate coordination, and no province is at the severe dissonance.

Fig 2 depicts the time-series trend of the synergy between HER and DIC at the nation and 4 regions from 2011 to 2020. The national-level analysis reveals an approximate "U"-shaped trend in the average CCD, while the overall synergy is at the moderate coordination. Among them, the average value of synergy development is the lowest in 2015 (0.4369), showing a slow

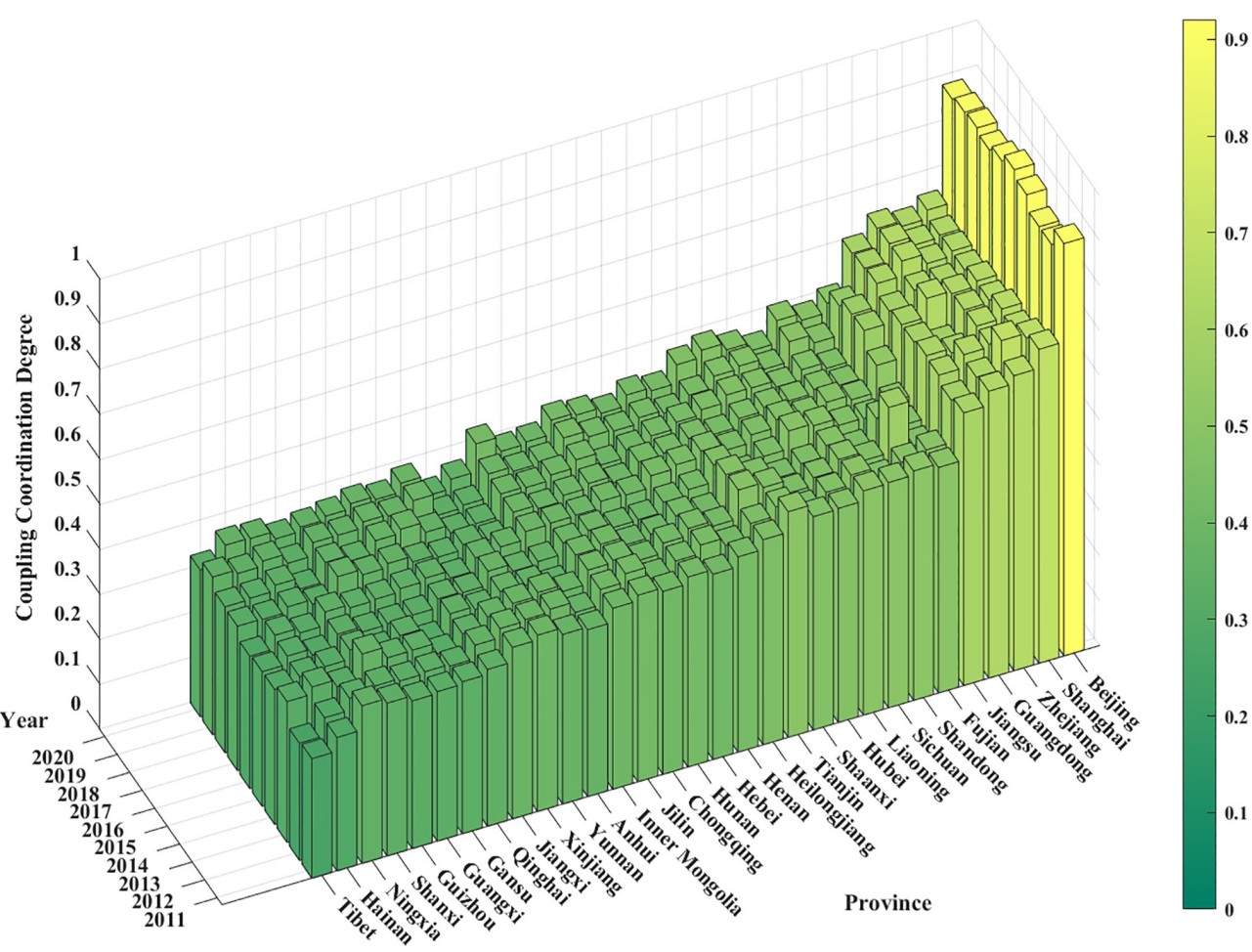

**Fig 1. Coupling coordination degree for each province in China from 2011 to 2020.**

downward trend before 2015 and a gentle upward trend after 2015. In terms of evolutionary trends, there are more pronounced differences in synergistic development among the four major regions. The East exhibits the greatest synergy, surpassing the national average significantly. However, there is a clear and steady decline in the gap between the East and the other three regions. The Central region exhibits a lower level of achievement compared to the national average, although there has been a fluctuating upward trend since 2014, resulting in a gradual reduction of the gap with the Northeast. The synergy in the West is the lowest, far lower than the national average, and showing a relatively obvious "U" trend. The government has effectively executed the higher education revitalization plan and the higher education basic capacity building project in central and western regions during the "13th Five-Year Plan" period. This has improved the regional distribution of educational resources and provided support to enhance the conditions for universities and colleges, resulting in the strengthening of high-quality higher education resources in central and western regions. China aims to bolster its supporting infrastructure, leverage the benefits of online teaching, and facilitate knowledge sharing of high-quality teaching resources, in order to assist universities and colleges in the central and western regions of the country to upscale their educational quality and reduce the gap with more developed regions during the 14th Five-Year Plan period. The synergy in

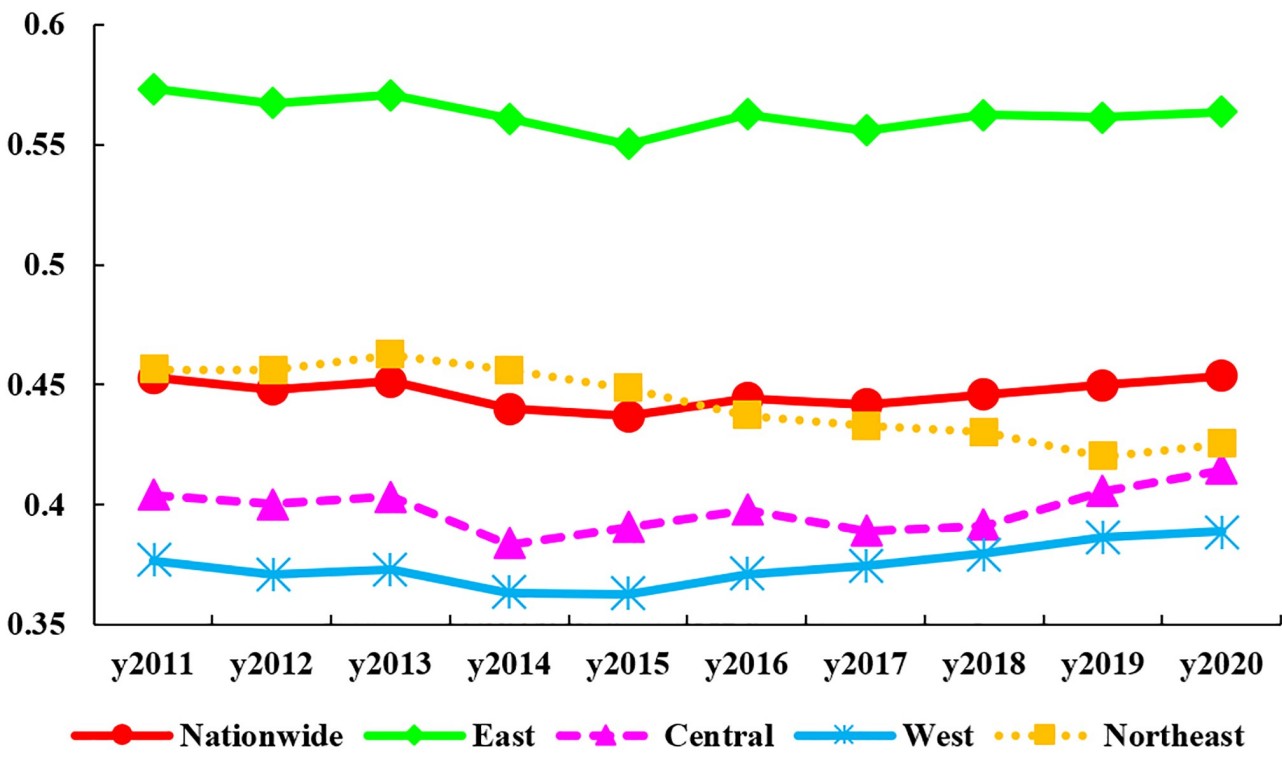

**Fig 2. Temporal trend of coupling coordination of HER and DIC.**

the Northeast is akin to the national standard but displays a gradual decline annually, dropping below the national mean post-2016.

## 5.2 Regional differences and sources of synergy between HER and DIC

Based on Eqs (9)–(14), this study derives the regional differences in the synergy of HER and DIC and analyses their overall differences, intra-regional and inter-regional differences, as well as the sources and contributions of the regional differences. The results are analyzed below.

Fig 3 depicts the trend of this synergistic development in the national and four regional Gini coefficients from 2011–2020. See Table 3 for details. The Gini coefficient for the country as a whole shows a fluctuating downward trend. Its value is 0.1522 in 2011 and drops to 0.1289 in 2020, a decrease of 15.31%, with an average annual decrease of 1.83%, which indicates that the imbalance of the synergy of HER and DIC across the country has been mitigated, and its regional difference is shrinking. From a regional perspective, the intra-regional difference in the East is the largest, with an average coefficient of 0.1424; the average values of the intra-regional differences in the Central and the West are closer, at 0.0615 and 0.0741 respectively; and the Northeast is the smallest (0.0339). This demonstrates that there are major differences in the synergies between the different provinces and cities in the East, which are very different from the other three regions, and which manifest themselves as spatial imbalances. Consistent with the nationwide trend, the coefficients for all four regions show a downward trend from 2011 to 2020. The downward process is smoothest in the East, where the coefficient falls from 0.1543 in 2011 to 0.1358 in 2020, a decline of 11.98%, with an average annual rate of decline of 1.41%. The Central's coefficient shows a trend of first decrease, then increase and then

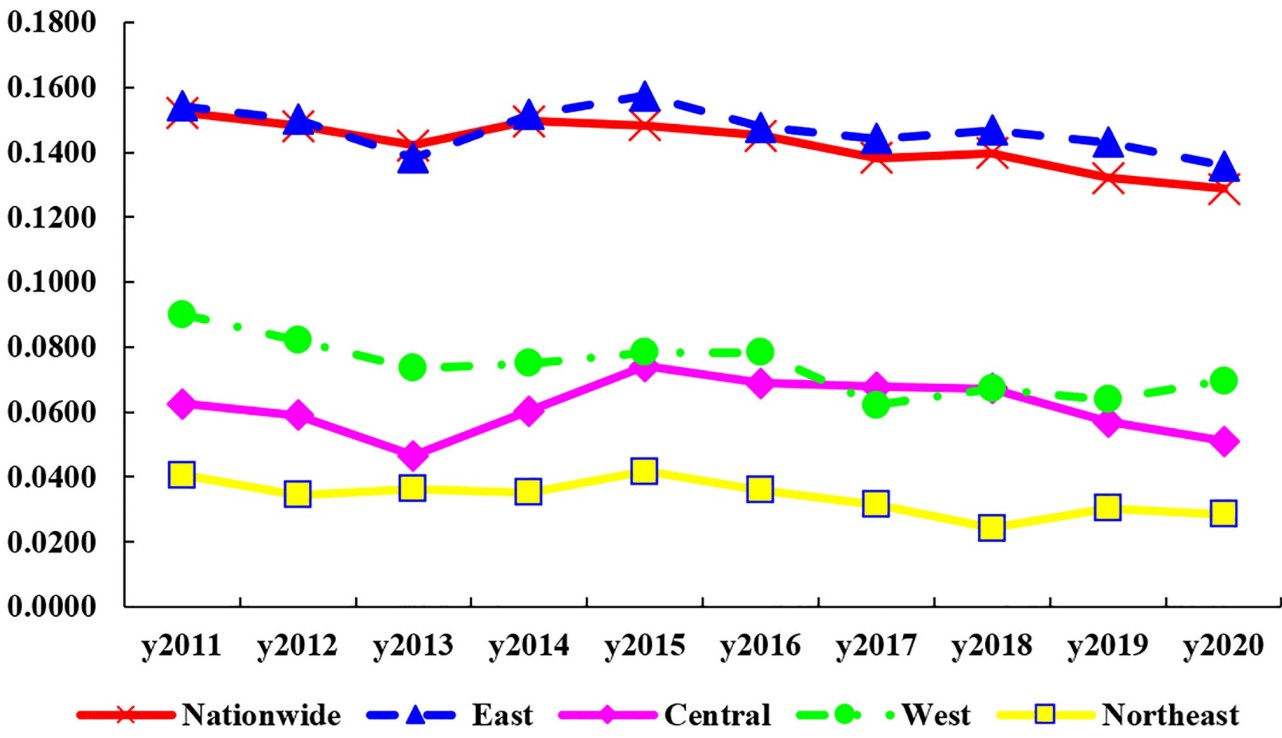

**Fig 3. Differences in intra-regional synergy of HER and DIC in China.**

decrease, with a rebound increase between 2013 and 2015, with values ranging from 0.0510 to 0.0625, a decrease of 18.48%. The measured values in the West ranged from 0.0697–0.0901, showing a fluctuating downward trend of 22.71%, but a rebound trend after 2019. The Northeast has the smallest intra-regional difference, with values ranging from 0.0285–0.0406, a decrease of 29.90%, indicating that its intra-regional difference tends to decrease further.

Fig 4 indicates the inter-regional differences and their evolution, and the results of the measurements are shown in Table 4. With the exception of the East-Northeast, the inter-regional differences show a fluctuating downward trend. The average value of the differences among

**Table 3. The Gini coefficient of CCD in China and four major regions.**

| Year | Nationwide | East | Central | West | Northeast |
|------|-----------|------|---------|------|-----------|
| 2011 | 0.1522 | 0.1543 | 0.0625 | 0.0901 | 0.0406 |
| 2012 | 0.1481 | 0.1502 | 0.0591 | 0.0821 | 0.0344 |
| 2013 | 0.1421 | 0.1382 | 0.0466 | 0.0736 | 0.0364 |
| 2014 | 0.1496 | 0.1516 | 0.0603 | 0.0750 | 0.0352 |
| 2015 | 0.1483 | 0.1575 | 0.0741 | 0.0784 | 0.0417 |
| 2016 | 0.1451 | 0.1478 | 0.0690 | 0.0783 | 0.0360 |
| 2017 | 0.1383 | 0.1442 | 0.0678 | 0.0622 | 0.0316 |
| 2018 | 0.1397 | 0.1469 | 0.0673 | 0.0673 | 0.0243 |
| 2019 | 0.1321 | 0.1432 | 0.0569 | 0.0639 | 0.0304 |
| 2020 | 0.1289 | 0.1358 | 0.0510 | 0.0697 | 0.0285 |

Note: Nationwide indicates the overall Gini coefficient. East, Central, West and Northeast denote intra-regional differences, respectively.

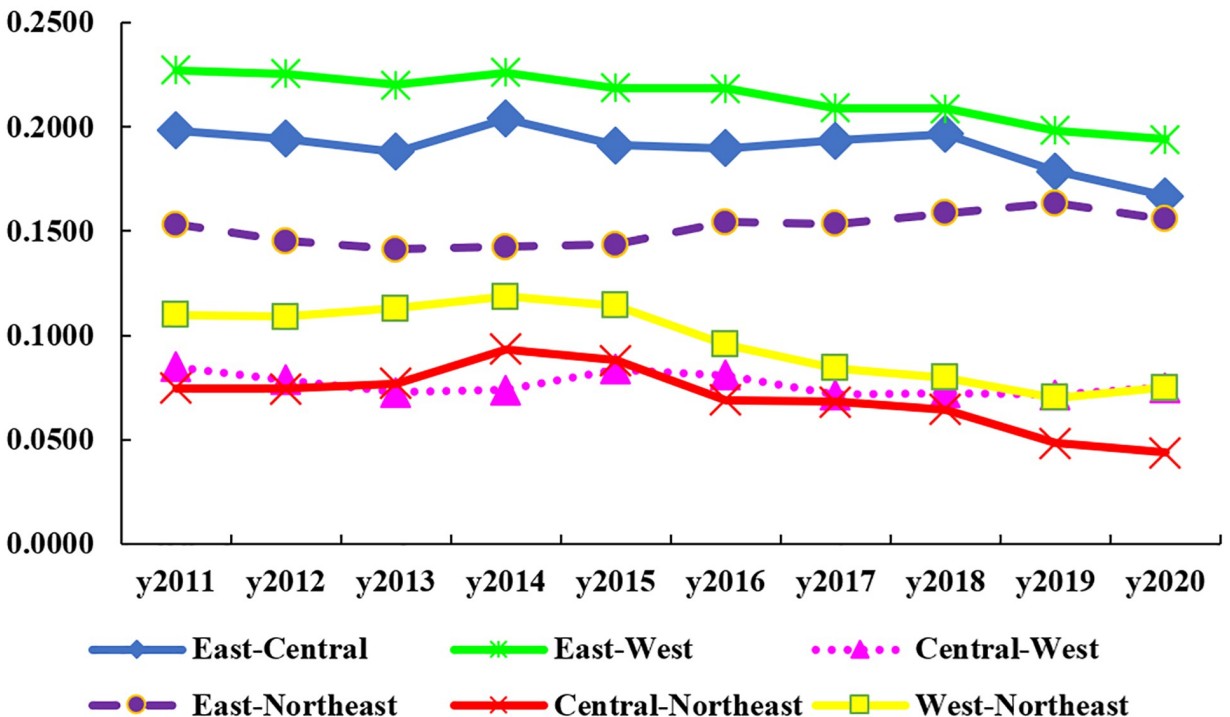

**Fig 4. Differences in inter-regional synergy of HER and DIC in China.**

synergy of HER and DIC is in the order of East-Central (0.1901), East-West (0.2146), East-Northeast (0.1511), Central-West (0.0765), Central-Northeast (0.0700), West-Northeast (0.0970). In 2020, the Gini coefficient of East-West is the largest (0.1940), followed by East-Central (0.1668), East-Northeast (0.1556), West-Northeast (0.0749), Central-West (0.0748), and Central- Northeast (0.0437). The gaps between the East and the other three areas are quite large. Among them, regional disparities appear to widen after 2013 and narrow again after 2019, with the Gini coefficient in the East-Northeast first decreasing, then increasing and then decreasing again. The difference is reduced by 31.80% and 41.34% respectively for the West-Northeast and Central-Northeast regions, which show an increasing and then decreasing trend. The difference of Central-West shows a few minor increases, but the general trend is

**Table 4. The inter-regional Gini coefficient in CCD of HER and DIC.**

| year | East-Central | East-West | Central-West | East-Northeast | Central-Northeast | West-Northeast |
|------|------|------|------|------|------|------|
| 2011 | 0.1982 | 0.2272 | 0.0849 | 0.1532 | 0.0745 | 0.1099 |
| 2012 | 0.1944 | 0.2256 | 0.0785 | 0.1452 | 0.0743 | 0.1088 |
| 2013 | 0.1880 | 0.2202 | 0.0727 | 0.1412 | 0.0769 | 0.1130 |
| 2014 | 0.2041 | 0.2260 | 0.0737 | 0.1423 | 0.0933 | 0.1185 |
| 2015 | 0.1911 | 0.2188 | 0.0838 | 0.1436 | 0.0881 | 0.1145 |
| 2016 | 0.1897 | 0.2183 | 0.0809 | 0.1544 | 0.0689 | 0.0958 |
| 2017 | 0.1935 | 0.2090 | 0.0715 | 0.1533 | 0.0681 | 0.0844 |
| 2018 | 0.1965 | 0.2087 | 0.0725 | 0.1585 | 0.0643 | 0.0799 |
| 2019 | 0.1786 | 0.1984 | 0.0715 | 0.1635 | 0.0484 | 0.0701 |
| 2020 | 0.1668 | 0.1940 | 0.0749 | 0.1556 | 0.0437 | 0.0749 |

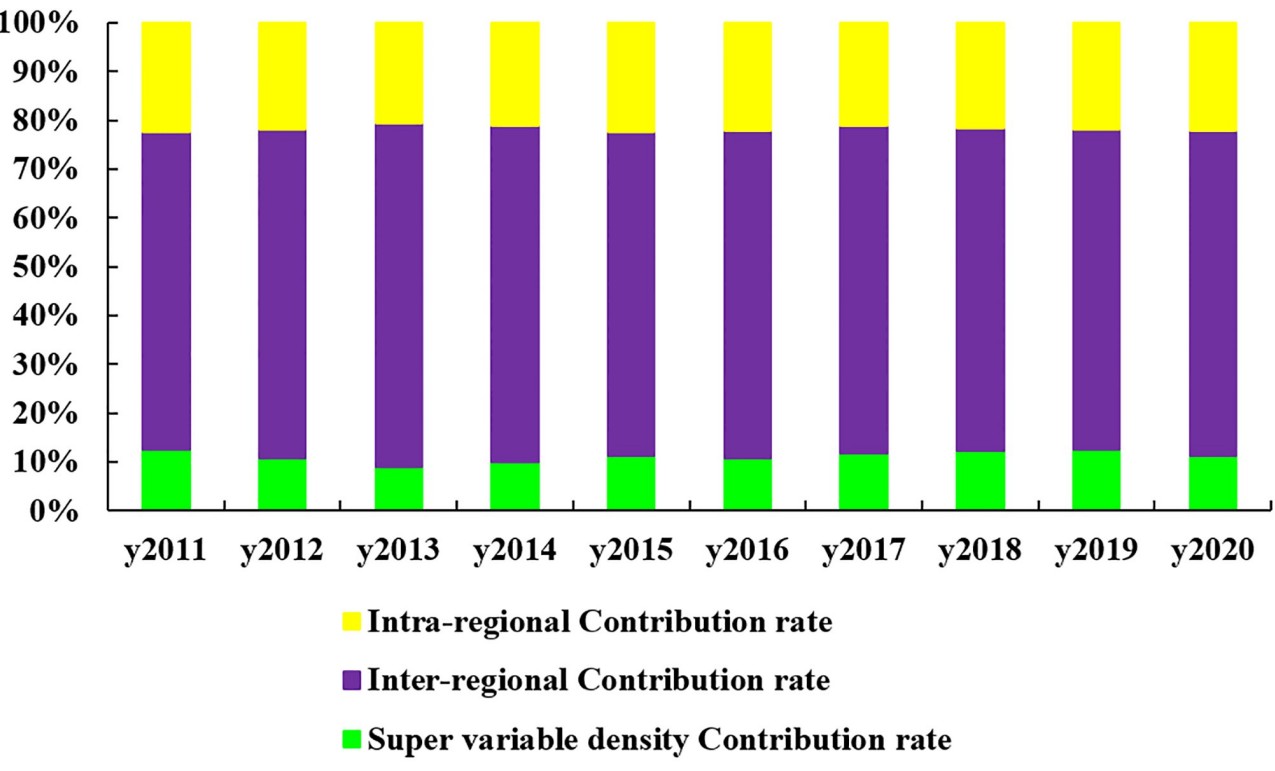

**Fig 5. Sources of overall differences in China's synergy of HER and DIC.**

downward. Generally, HER and DIC tend to harmonize across regions, with the gap between China's four main regions gradually narrowing.

The sources and contributions to the overall differences in the synergy between HER and DIC are shown in Fig 5, and the corresponding data are presented in Table 5. In relation to the impact of regional variations, it can be observed that the largest proportion of differences is attributed to inter-regional disparities. These differences range from 64.98% to 70.50%, with an average contribution rate of 67.07%. Notably, there is a pattern of initially increasing and

**Table 5. Source and contribution rate of regional differences in synergy of HER and DIC.**

| year | Intra-regional differences | | Inter-regional differences | | Super variable density | |
|---|---|---|---|---|---|---|
| | Gw | Contribution rate (%) | Gnb | Contribution rate (%) | Gt | Contribution rate (%) |
| 2011 | 0.0340 | 22.35 | 0.0989 | 64.98 | 0.0193 | 12.67 |
| 2012 | 0.0323 | 21.79 | 0.1000 | 67.53 | 0.0158 | 10.68 |
| 2013 | 0.0292 | 20.56 | 0.1002 | 70.50 | 0.0127 | 8.94 |
| 2014 | 0.0317 | 21.19 | 0.1031 | 68.91 | 0.0148 | 9.89 |
| 2015 | 0.0333 | 22.44 | 0.0983 | 66.33 | 0.0167 | 11.23 |
| 2016 | 0.0319 | 22.00 | 0.0975 | 67.19 | 0.0157 | 10.81 |
| 2017 | 0.0293 | 21.20 | 0.0928 | 67.08 | 0.0162 | 11.72 |
| 2018 | 0.0303 | 21.69 | 0.0924 | 66.12 | 0.0170 | 12.19 |
| 2019 | 0.0290 | 21.96 | 0.0866 | 65.54 | 0.0165 | 12.50 |
| 2020 | 0.0285 | 22.12 | 0.0858 | 66.56 | 0.0146 | 11.33 |
| Mean | 0.0310 | 21.73 | 0.0955 | 67.07 | 0.0159 | 11.20 |

subsequently decreasing inter-regional differences. This trend suggests that inter-regional disparities have emerged as the primary factor influencing the overall differences in the synergy of HER and DIC. Intra-regional differences range from 20.56% to 22.44%, with the average rate being 21.73%. The contribution of intra-regional differences is small in comparison with inter-regional differences and is not the main cause of regional differences. The pattern observed in the super variable density follows a similar trend as the intra-regional differences. Initially, both decrease, then gradually increase, and eventually stabilize. It indicates that the cross-over of the inter-regional sample does not have a significant impact on the regional differences, as the contribution ranges from 8.94% to 12.67%, with an average contribution of 11.20%. Its increase indicates that certain provinces in the eastern region, which have less synergy, will exhibit a lower CCD compared to the more developed provinces in the West. This observation highlights the effectiveness of the government's efforts to enhance the foundational capabilities of universities in the western regions and to augment the availability of high-quality educational resources in those areas. Hence, to address the issue of regional disparities in their synergy, it is imperative to prioritize the advancement of China's HER and DIC in a manner that fosters greater coordination and sustainability, with the goal of bridging the gap among different regions.

## 5.3 Dynamic evolution of HER and DIC synergies

To uncover the distributional dynamics and evolutionary patterns of the two subsystems' synergy, this study thoroughly analyzes the absolute differences within the nationwide and four major regions through kernel density estimation. The analysis covers distributional locations, dynamics, extensibility, and polarization trends.

**5.3.1 Analysis of nationwide's kernel density estimation.** Fig 6 demonstrates how the synergy between HER and DIC in China has developed over the period from 2011 to 2020. During the observation period, the overall kernel density function underwent a leftward shift

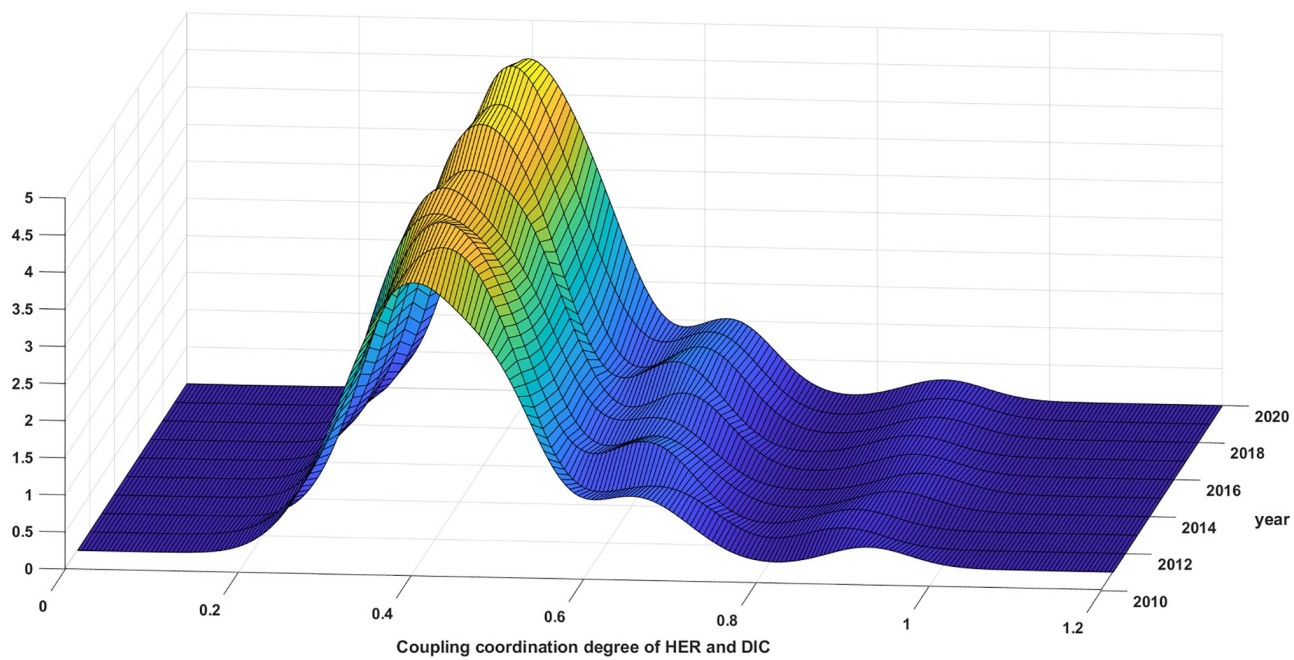

**Fig 6. Distribution of synergy between HER and DIC in China.**

followed by a rightward shift in distributional position. This suggests that the synergy in nationwide first decreased and then increased, in line with the typical findings presented in the previous section. The distribution curve indicates a decreasing trend in the absolute difference of this synergy. This is evident from the increasing height of the main peak and the narrowing width of the curve. Regarding distribution extensibility, the curve of the distribution displays a notable right skew, with its extensibility expanding. This signifies that provinces experiencing synergy are maintaining a robust trajectory, which is exacerbating the discrepancy with other provincial regions. Regarding polarization trends, the distribution curve has one primary peak and several sub-peaks, with the latter being lower. This indicates a weak form of multi-polarization, and suggests that provincial development has a gradient effect due to digitalization and regional economic growth. On the one hand, as the primary provider of higher education resources, the regional economic development and financial payment capacity to which it belongs profoundly affects the layout of regional higher education resources through such elements as funding inputs, school running conditions, and faculty allocation. On the other hand, the higher the regional digitalization means the more complete the digital infrastructure construction, and the universities are more active in using digital technology for teaching and research. Regions such as Beijing-Tianjin-Hebei, the Yangtze River Delta, and the Pearl River Delta, which depend on robust economic development advantages and well-established digital infrastructure, can successfully merge higher education resources with digital technology. This integration is more likely to enhance the quality of teaching and scientific research. However, the central and western provinces may face challenges in catching up with such advancements in the near future.

**5.3.2 Analysis of regional kernel density estimation.** The distributional dynamics and evolutionary trends of the synergy between HER and DIC in the 4 major regions from 2011 to 2020 are illustrated in Figs 7–10. The distribution curve in the East exhibits a variable leftward shift with regards to the displacement of the primary peak. This suggests that the CCD in the

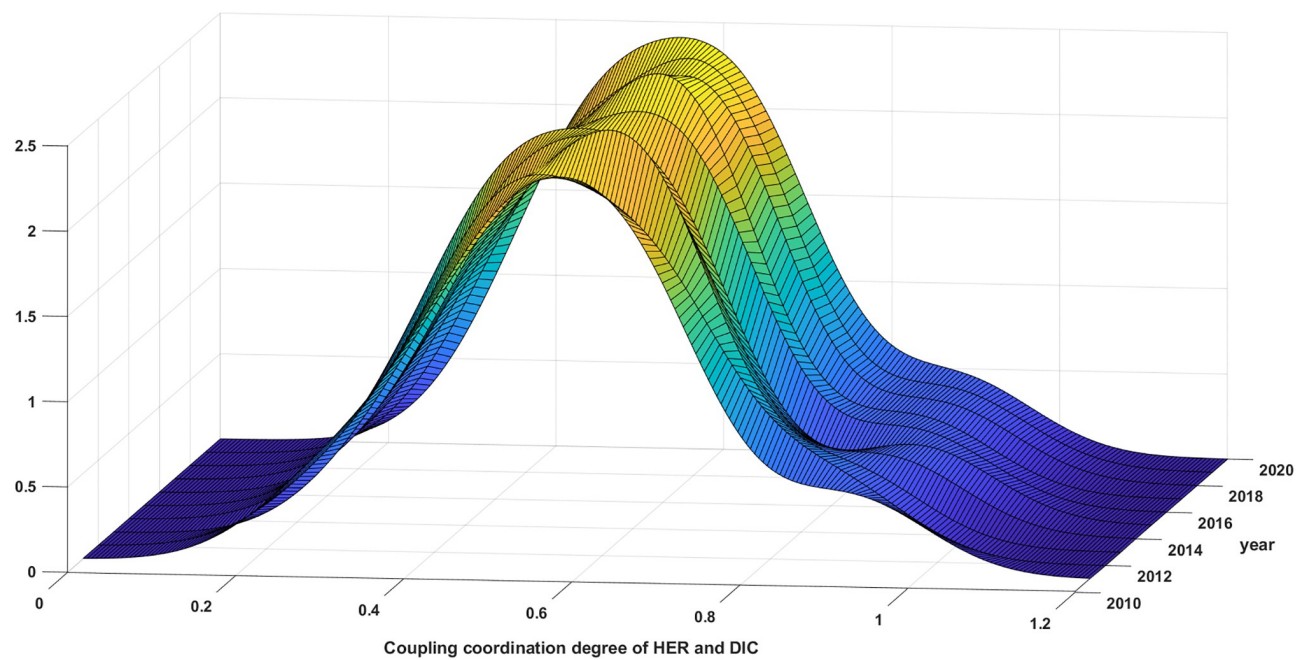

**Fig 7. Distribution of synergy between HER and DIC in the eastern region.**

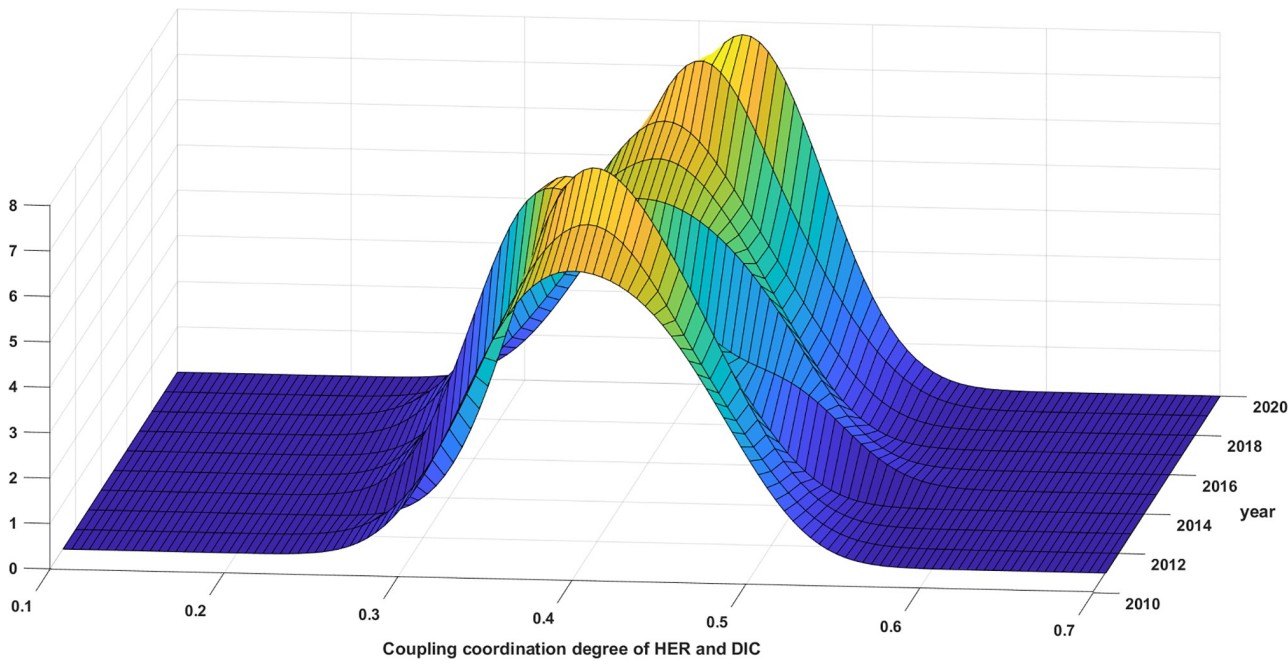

**Fig 8. Distribution of synergy between HER and DIC in the central region.**

Eastern region displays a propensity to decrease throughout the duration of the observed samples. See Fig 7 for details. The elevation of the primary summit is on the rise, the breadth of the curve is contracting, and the expansion of the distribution is not readily apparent. These observations suggest a decline in the absolute disparity within the eastern region, with the inter-

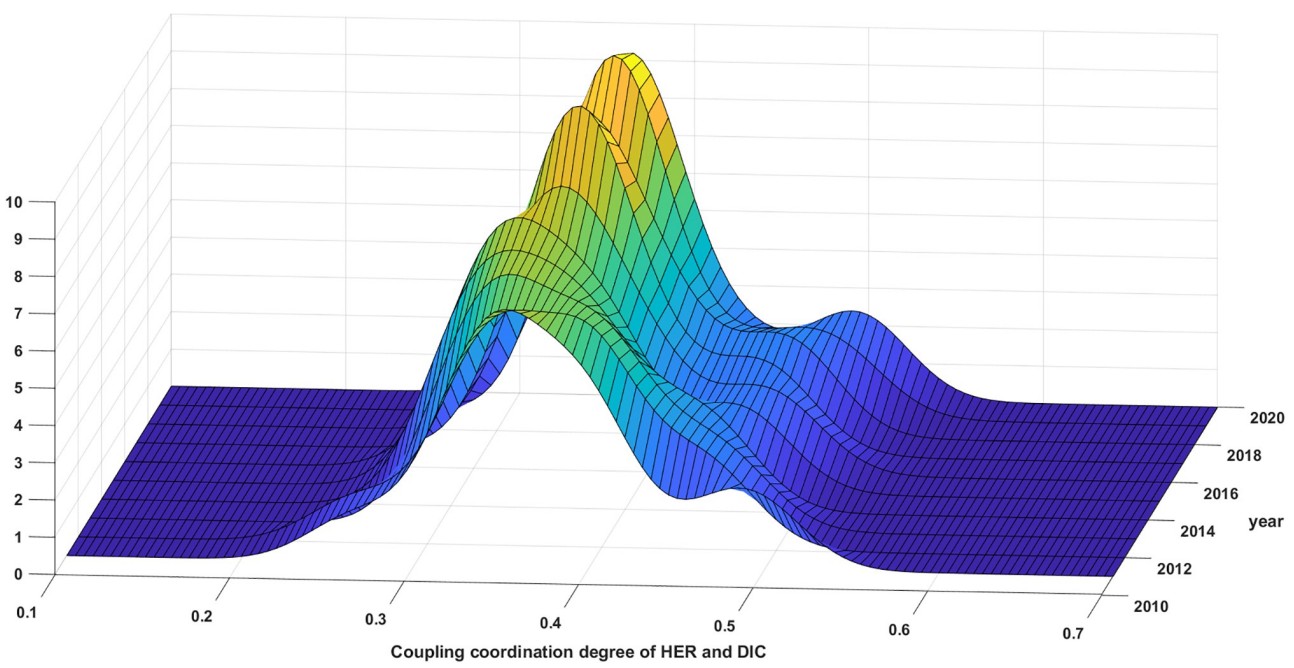

**Fig 9. Distribution of synergy between HER and DIC in the western region.**

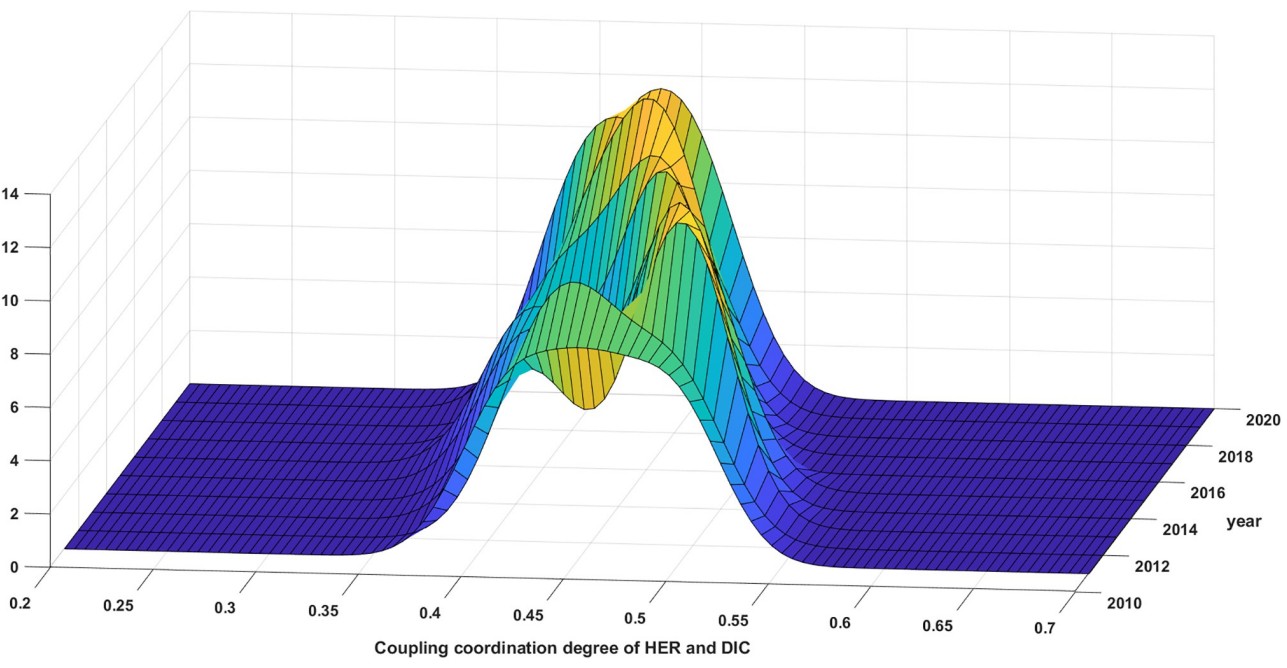

**Fig 10. Distribution of synergy of HER and DIC in the northeast region.**

provincial discrepancy gradually diminishing. Meanwhile, there is only one primary peak, signifying the absence of polarization.

According to Fig 8, the distribution curve in the Central exhibits a noteworthy pattern whereby the main peak shifts left and then right. This overall trend is characterized by a rightward shift and an increase in the height of the main peak. Furthermore, the distribution curve consistently displays a right trailing and widening extension, without any indication of regional polarization. These findings align with the earlier analysis of the Gini coefficient, which suggests that the synergy between HER and DIC in the central region is progressively strengthening. Additionally, there is a tendency for the absolute difference to diminish.

The synergy between HER and DIC in the West is depicted in Fig 9, showcasing the distribution dynamics and evolution trend. The distribution curve demonstrates a shift to the left followed by a shift to the right, while the height of the primary peak exhibits a gradual increase. These observations suggest a general upward trend in the synergistic development of the western region. Throughout the study period, an intriguing transformation in the primary peak's shape from a "flat and wide" structure to a "sharp and narrow" configuration was observed. Remarkably, this prominent peak consistently comprised both primary and side peaks. It is noteworthy, however, that the side peaks were consistently lower in elevation, suggesting the presence of a gradient effect in the western region's synergy. This phenomenon also indicates a discernible polarization trend within the studied area. Meanwhile, there has been an expansion in the position of the primary peak as well as the side peaks. Additionally, the height of the side peaks has shown an increase, suggesting a growing polarization in this collaborative progress. Moreover, it is evident that there is a tendency for the absolute differences to amplify in the western region.

Fig 10 illustrates that the primary peak of the distribution curve in the Northeast has predominantly shifted towards the left. Furthermore, there has been a slight increase in the overall height of the primary peak, albeit with certain years, such as 2019, experiencing a decline.

Moreover, the width of the primary peak has displayed a minor reduction, while the right trailing has remained relatively stable without significant alterations. Notably, the presence of a solitary primary peak, devoid of regional polarization, indicates a decline in the synergy of the HER and DIC in the northeast region. However, it is worth noting that the absolute difference between these elements is diminishing.

## 5.4 Trend forecasting and spatial impact of synergy between HER and DIC

**5.4.1 Analysis of traditional markov chain.** To anticipate the future trajectory of the interplay between HER and DIC, this study employs the conventional Markov chain and spatial Markov chain for examination. By adhering to the quartile principle of coupling coordination, the 31 provinces in China are categorized into four distinct types: low level (L), medium-low level (ML), medium-high level (MH) and high level (H). The transfer probability matrix is subsequently computed under the condition of a 1-year time lag, and the results are displayed in Table 6.

The transfer probability matrix of a conventional Markov chain demonstrates a notable stability in the transfer probabilities across all chain orders. Without the effect of hysteresis, the probabilities of no transfer are 83.78%, 79.10%, 82.86% and 88.24% for L, ML, MH and H. The probabilities of downward transfer for ML, MH and H are 11.94%, 7.14% and 11.76%. Upward transfer probabilities for L, ML and MH are 16.22%, 8.96% and 10.00%. The likelihood of an adverse transition occurring for MH is 0%, while the probability of a transfer from H to ML and L is also 0%. Observations reveal that the probabilities along the diagonal of the matrix surpass those on the non-diagonal, thereby signifying the presence of club convergence characteristics in the synergy of HER and DIC. Moreover, the substantial level of convergence accentuates the existence of a multi-polarization in the said synergy development. Convergent clubs mostly occur between neighboring types, and the probability of cross-state transfer occurring is non-existent. The provinces with MH exhibit a higher likelihood of upward transfer, suggesting a notable inclination towards synergy in these regions. Conversely, the provinces with ML showcase a lower probability of upward transfer in comparison to downward transfer, indicating a declining trend of convergence development in these particular areas.

**5.4.2 Analysis of spatial markov chain.** To further investigate the impact of spatial proximity on local synergy transfer, the analysis in this work employs the spatial Markov chain approach. We classify the neighboring areas in a province into four spatial lag types based on the ratio of the average CCD of neighboring areas in a province to the national average CCD. Adjacent areas with this ratio below 25% are spatial lag type I, areas between 25% and 50% are spatial lag type II, areas between 50% and 75% are spatial lag type III, and areas greater than 75% are spatial lag type IV. Similarly, the transfer probability matrix is computed conditional on a 1-year lag. See Table 7 for details.

The result presented in Table 7 demonstrates that the diagonal probability does not consistently surpass the non-diagonal probability within the spatial transfer probability matrix. This suggests the potential occurrence of "jumping transfer" of certain synergistic developments

**Table 6. Transfer probability matrix for synergy of HER and DIC.**

| Type | L | ML | MH | H |
|------|------|------|------|------|
| L | 0.8378 | 0.1622 | 0.0000 | 0.0000 |
| ML | 0.1194 | 0.7910 | 0.0896 | 0.0000 |
| MH | 0.0000 | 0.0714 | 0.8286 | 0.1000 |
| H | 0.0000 | 0.0000 | 0.1176 | 0.8824 |

**Table 7. Spatial transfer probability matrix for synergy of HER and DIC.**

| Type of space | Type | L | ML | MH | H |
|---|---|---|---|---|---|
| I | L | 1.0000 | 0.0000 | 0.0000 | 0.0000 |
| | ML | 0.1000 | 0.9000 | 0.0000 | 0.0000 |
| | MH | 0.0000 | 0.0000 | 0.5000 | 0.5000 |
| | H | 0.0000 | 0.0000 | 0.0000 | 0.0000 |
| II | L | 0.8372 | 0.1628 | 0.0000 | 0.0000 |
| | ML | 0.1818 | 0.8182 | 0.0000 | 0.0000 |
| | MH | 0.0000 | 0.0830 | 0.7867 | 0.1303 |
| | H | 0.0000 | 0.0000 | 0.2778 | 0.7222 |
| III | L | 0.8667 | 0.1333 | 0.0000 | 0.0000 |
| | ML | 0.0526 | 0.7895 | 0.1579 | 0.0000 |
| | MH | 0.0000 | 0.1304 | 0.8261 | 0.0435 |
| | H | 0.0000 | 0.0000 | 0.0667 | 0.9333 |
| IV | L | 0.8000 | 0.2000 | 0.0000 | 0.0000 |
| | ML | 0.1250 | 0.6875 | 0.1875 | 0.0000 |
| | MH | 0.0000 | 0.1333 | 0.8000 | 0.0667 |
| | H | 0.0000 | 0.0000 | 0.0571 | 0.9429 |

following the consideration of spatial factors. Specifically, the stability of provinces at the L-level decreases and the likelihood of upward transfer to ML increases as the spatial lag type increases. This effect is particularly pronounced when the spatial lag type is IV, with a 20% probability of upward transfer. This probability is higher than that of the traditional Markov chain (16.32%), indicating that high-level neighborhoods play a significant role in facilitating the transfer of local synergy.

When the spatial lag types are categorized as III and IV, there is a noticeable trend of decreasing stability in ML level provinces. Additionally, the upward transfer probabilities in these cases are observed to be 15.79% and 18.75% respectively, which are significantly larger than the traditional Markov chain value of 8.96%. These findings suggest that the presence of middle-high and high-level neighborhoods positively contribute to the enhancement of local synergy. The provinces with MH level are experiencing a decrease in stability as the spatial lag type increases. However, the likelihood of an upward shift is declining. In the case of spatial lag type II, the probability of a downward shift is 8.30%, surpassing the traditional Markov chain probability of 7.14%. This suggests that neighboring provinces with low and medium levels may hinder the local synergy. The stability of provinces categorized as H-level exhibits a consistent upward trend, with a probability of 94.29% when considering the spatial lag type as IV. However, the presence of lower-level neighborhoods has a detrimental effect, causing a downward shift in stability with a probability of 27.78% when the neighborhood is categorized as II. This probability is higher than the 11.76% probability when not taking into account the spatial factor. Furthermore, when the spatial lag type is II, the probability of a downward shift remains at 27.78%, exceeding the 11.76% probability without considering the spatial factor.

Spatial factors play a crucial role in the dynamic transfer of synergy between HER and DIC in China. Lower-level neighborhoods tend to hinder the local coordination between these two subsystems, thus impeding their synergistic development. In contrast, medium and high-level neighborhoods contribute positively to the local synergistic development. This could be attributed to the comparative advantages of high-level neighborhoods in terms of teaching resources and digitization. These advantages facilitate the establishment of digital education collaborative zones and foster inter-regional exchanges and cooperation. Consequently, high-level

neighborhoods act as catalysts, promoting the local synergistic development of HER and DIC and elevating the overall quality of higher education endeavors.

## 6. Discussion

To elucidate the synergistic interaction between HER and DIC, we compiled panel data from China's 31 provinces from 2011 to 2020. We employed the CCD model to measure their synergistic progress and utilized the Dagum Gini coefficient, kernel density estimation, traditional Markov chain, and spatial Markov chain methods to analyze regional disparities, temporal changes, and predictive trends of this synergy. We observed that the synergy between HER and DIC is moderate coordination, with good coordination provinces predominantly in the east, whereas the majority in the west are at a primary coordination. Notably, the eastern region exhibits the highest average level of synergy development. In contrast, the northeastern region, while on par with the national average, has been declining annually. Both the central and western regions lag behind the national average, yet they have each shown an upward trajectory since 2014. The observed regional disparities correlate with the uneven distribution of higher education resources in China [13], as well as the variability in the digital economy's development among different regions [15, 16]. The Dagum Gini coefficient demonstrates significant regional disparities in synergistic development, with inter-regional differences accounting for 67.07% of these disparities. The East-West divide exhibits the most pronounced differences, while the Central-Northeast divide is the least significant. Yet, the overall Gini coefficient exhibits a declining trend, suggesting that the disparities in regional synergistic development are diminishing. The results of this study are consistent with those reported by Shi [50]. The kernel density estimation confirms a reduction in the CCD imbalance. Specifically, the eastern region's CCD is moving leftward, with a reduction in absolute differences and a single peak, indicating no polarization. Synergistic development in the central region is increasing, with diminishing absolute differences and no polarization evident. The western region's trend fluctuates, initially decreasing before rising, consistently accompanied by a side peak and a gradient effect, indicating a polarization tendency. The north-eastern region's development is declining, yet absolute differences are reducing, with no evidence of polarization. Markov chain predictions indicate that synergistic development primarily transitions between adjacent types, which complicates achieving a significant leap in development. Middle and high-level neighborhoods enhance local synergistic development, while lower-level neighborhoods may impede progress and potentially reverse gains.

## 7. Conclusions and policy implications

### 7.1 Conclusions

The integration of HER and DIC is vital to China's educational reform, and addressing regional disparities in their combined advancement is key to improving higher education quality. The key findings of this study are as follows: (1) The coordinated development of HER and DIC exhibits a fluctuating trend, initially declining then rising, with the eastern region demonstrating notably higher synergy compared to the other three, highlighting pronounced regional disparities. (2) The synergistic development of HER and DIC reveals regional disparities, with the eastern region having the greatest intra-regional difference and the most significant inter-regional difference occurring between East-West, primarily attributed to inter-regional differences. (3) While regional disparities in the synergistic development are diminishing, a weak trend toward multi-polarity is emerging, with polarization being particularly evident in the western region. (4) The synergistic development exhibits four distinct patterns of convergence,

with the most pronounced convergence levels being of greater significance, and spatial factors exerting a considerable influence on regional synergy.

### 7.2 Policy implications

Drawing upon the aforementioned findings, this study proposes the subsequent suggestions in order to optimize the integration of HER and DIC, while simultaneously mitigating regional disparities, with the ultimate aim of bolstering the overall caliber of higher education in China.

(1) To enhance the synergy between HER and DIC, it is essential for all regions to bolster the informatization of higher education and ensure the smooth integration of digitalization within it. The education sector should promote innovative teaching methods, including online and distance learning. Institutions should develop digital platforms for course selection, library access, and student information management to expedite digital transformation. Conversely, government departments should consistently expand digital infrastructure and encourage corporate R&D in digital technology. This aims to further foster digital transformation and innovation. The integration of advanced IT, including big data, AI, and blockchain, with higher education resources will create a new ecosystem for digital transformation in academia. This integration will harness new digital technologies to steer higher education into a new era of digitalization.

(2) Active exploration is underway for regional cooperation mechanisms to foster the synergistic development of HER and DIC. To redress spatial imbalances in the concurrent development of HER and DIC, each province must craft tailored strategies to reduce regional disparities. Among these strategies, leveraging the eastern region's inherent strengths is crucial. This objective can be realized by creating an interconnected spatial network to promote the free flow of capital, technology, and talent among regions, thereby fostering a new synergistic development model conducive to the modernization of China's educational sector.

(3) Regions with lower synergistic development levels should learn from the advanced experiences of more developed provinces. This will enhance the effective integration of HER with DIC, reducing regional disparities and leveraging the spatial linkage effect for synergistic development. Thus, local governments must dismantle administrative and spatial barriers to facilitate inter-regional communication and establish a harmonized and mutually reinforcing framework, thereby stimulating high-level provinces to lead their neighbors in development and elevating the national standard of synergistic development, which will ultimately foster sustainable growth in China's higher education sector.

## 8. Contributions and limitations

This study offers insights into addressing disparities in digital higher education across regions. Distinct from other research on higher education and digitization, this study contributes to the digital education literature by: elucidating the integration mechanism between HER and DIC, assessing the synergistic development through the CCD model, and examining regional disparities, spatial distribution, and forecasting trends in this synergistic development, thereby establishing a theoretical foundation for subsequent studies and offering empirical evidence to address the imbalance in China's higher education sector. Nevertheless, the study has certain limitations: Firstly, as a provincial study, it suggests that future research could collect urban data for a more granular analysis of higher education's digital transformation. Secondly, the

study has not yet investigated the driving factors behind this synergistic development; future research should address this omission.

## Supporting information

**S1 Data.**
(XLSX)

## Acknowledgments

The authors would like to thank the anonymous reviewers for their valuable comments on drafts of this paper.

## Author Contributions

**Conceptualization:** Ying Xie.

**Data curation:** Ying Xie.

**Formal analysis:** Ying Xie, Minglong Zhang.

**Investigation:** Ying Xie.

**Methodology:** Minglong Zhang.

**Software:** Minglong Zhang.

**Supervision:** Ying Xie, Minglong Zhang.

**Visualization:** Minglong Zhang.

**Writing – original draft:** Ying Xie.

**Writing – review & editing:** Minglong Zhang.

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
