## [Decision Letter · Decision Letter 0]

19 Mar 2024

PONE-D-24-03628Synergy of higher education resources and digital infrastructure construction in China: regional differences, dynamic evolution and trend forecastingPLOS ONE

Dear Dr. Zhang,

Thank you for submitting your manuscript to PLOS ONE. After careful consideration, we feel that it has merit but does not fully meet PLOS ONE’s publication criteria as it currently stands. Therefore, we invite you to submit a revised version of the manuscript that addresses the points raised during the review process.

We look forward to receiving your revised manuscript.

Kind regards,

Muhammad Farooq Umer, PhD Epidemiology and Health Statistics

Academic Editor

PLOS ONE

Journal Requirements:

**Additional Editor Comments:**

This manuscript needs a strengthened discussion section and some changes in the introduction section. Furthermore, one of the reviewers has recommended that you cite two published articles, which you may decide to include or not, depending on their suitability. You may include the recommended ones or similar articles in your study as a reference. 

Reviewers' comments:

Reviewer's Responses to Questions

**Comments to the Author**

1. Is the manuscript technically sound, and do the data support the conclusions?

Reviewer #1: Yes

Reviewer #2: Yes

2. Has the statistical analysis been performed appropriately and rigorously? 

Reviewer #1: Yes

Reviewer #2: Yes

3. Have the authors made all data underlying the findings in their manuscript fully available?

Reviewer #1: Yes

Reviewer #2: Yes

4. Is the manuscript presented in an intelligible fashion and written in standard English?

Reviewer #1: Yes

Reviewer #2: Yes

5. Review Comments to the Author

Reviewer #1: This study establishes two comprehensive evaluation frameworks for higher education resources (HER) and digital infrastructure construction (DIC), using panel data from all provinces spanning the period from 2011 to 2020. The comments are as follows:

1. The study focuses on the Mainland of China. Thus, it is better for authors to revise “China” to “Mainland of China” is appropriate places (e.g., selection of data in the methods part) in the manuscript.

2. Is it possible to shorten the length of the Introduction part. The words look very redundant.

3. After the description of results and before the conclusion, the manuscript needs an in-depth discussion, explaining the mechanism of the results in the study and compare with other published studies, using possible references. Otherwise, readers may think that the explanation may just come from the authors’ own imagination but not from scientific evidence. The authors can revise and improve the writing in the results part with added references; or write discussion in a separate part (recommended).

4. The strengths and limitations of the study should be specially addressed, e.g., using sub-headings to address.

5. The conclusions and implications should be shortened and made more concise, some words and explanations should be relocated to discussion part.

6. The manuscript requires a language revision.

Reviewer #2: The study titled “Synergy of Higher Education Resources and Digital Infrastructure Construction in China: Regional Differences, Dynamic Evolution and Trend Forecasting.” However, it needs a few minor revisions to improve its quality. My details comments are as below.

• The abstract should present the research objectives, methods, and concise results of the study. Revise these concerns.

• The introduction section is nicely developed. Mention the objectives in this section.

• Rewrite the following paragraph in the introduction section for more clearance” Since the beginning of the century, an increasing number of universities have been developing "digital education" with the help of ICT [19]. A richer exploration of the link between higher education and digitization has been undertaken by scholars, including the following aspects: Firstly, the need for digital technologies in higher education. A multitude of academic investigations has consistently substantiated the fact that the COVID-19 pandemic has necessitated higher education establishments to adopt online learning platforms as a means to uphold their educational frameworks. This situation has underscored the critical importance of integrating digital technologies into the global education system [20-22]. Furthermore, to maximize student engagement in the classroom, Penprase (2018) contended that incorporating projectors, computers, and other advanced technological devices in higher education is likely to captivate students' interest, thereby promoting their active participation in classroom learning and offering them an immersive learning experience [23]. Additionally, Ozdamli and Cavus (2021) emphasized the significance of feedback loops in digital classrooms, as they enable students to receive timely feedback from their instructors in real time [24].

• Add the following latest literature on the topic

• https://doi.org/10.1371/journal.pone.0295979

• https://doi.org/10.1371/journal.pone.0294902

• Check the table numbers.

• Page number is missing please add

• Results should be backed by literature. Add more discussion and add citations to back your results.

• Add concise and comprehensive study results and more policy recommendations in the conclusion section.

• Avoid grammatical errors throughout the manuscript.

6. PLOS authors have the option to publish the peer review history of their article (what does this mean?). If published, this will include your full peer review and any attached files.

Reviewer #1: No

Reviewer #2: No

---

## [Author Response · Author response to Decision Letter 0]

1 May 2024

Dear Editor and Reviewers:

Thank you for your letter and for the reviewers’ comments concerning our manuscript entitled “Synergy of higher education resources and digital infrastructure construction in China: regional differences, dynamic evolution and trend forecasting” (PONE-D-24-03628). Those comments are all valuable and very helpful for revising and improving our paper, as well as the important guiding significance to our researches. We have studied comments carefully and have made correction which we hope meet with approval. Revised portion are marked with track changes. The main corrections in the paper and the responds to the reviewers’ comments are as following: 

Responds to the reviewer’s comments:

Reviewer #1: This study establishes two comprehensive evaluation frameworks for higher education resources (HER) and digital infrastructure construction (DIC), using panel data from all provinces spanning the period from 2011 to 2020. The comments are as follows:

1. The study focuses on the Mainland of China. Thus, it is better for authors to revise “China” to “Mainland of China” is appropriate places (e.g., selection of data in the methods part) in the manuscript.

Response: Thanks for the reviewer’s kind reminder. Considering the reviewer’s comment, we have modified this part of text expression. The precedent version of this text expression has been replaced, becoming “31 provinces in mainland China”. Please read the Section 4.3 Data Sources, on Page 11.

2. Is it possible to shorten the length of the Introduction part. The words look very redundant.

Response: Just like what the reviewer said, the Introduction part is very redundant. We have modified the Introduction part of text expression according to reviewer’s comment. The precedent version of this text expression has been replaced, becoming “The COVID-19 pandemic in early 2020 significantly impacted higher education [1]. Several international organizations, including the United Nations Educational, Scientific and Cultural Organization (UNESCO), the United Nations (UN), the Organization for Economic Co-operation and Development (OECD), and the World Bank, proposed recommendations advocating for the advancement of information and communication technology (ICT) and virtual educational models [2-5], and promoting digital environments in higher education [6], to mitigate the negative impacts of the pandemic. Meanwhile, The International Association of Universities (IAU) had recently released a publication entitled "Transforming Higher Education in a Digital World for the Global Common Good." This publication advocates for a concentrated effort in digitizing higher education, with a specific emphasis on ethical considerations, inclusivity, and the pursuit of initiatives that prioritize the welfare of the global community. It is essential that these goals are achieved through the training provided [7]. EDUCAUSE published Top 10 IT Issues, 2020: The Drive to Digital Transformation Begins, describing the key issues driving digital transformation in higher education [8]. In August 2021, the Ministry of Education in China gave its approval for Shanghai to serve as a pilot zone for the transformation of digital education [9]. Subsequently, on January 16-17, 2022, a national education conference was held to implement the initiative for digital education strategy [10]. This series of related policies has catalyzed the demand for digital construction in higher education, accelerated the integration of higher education and digital technology, and thus has become a new hotspot that has attracted much attention in the field of higher education, which will surely become a major breakthrough in the reform and development of higher education in China.”. Please read 3 paragraphs content in the Introduction, on Page 1-2.

3. After the description of results and before the conclusion, the manuscript needs an in-depth discussion, explaining the mechanism of the results in the study and compare with other published studies, using possible references. Otherwise, readers may think that the explanation may just come from the authors’ own imagination but not from scientific evidence. The authors can revise and improve the writing in the results part with added references; or write discussion in a separate part (recommended).

Response: Thank you for the suggestion. We have added the discussion required as explained above. Please read the Section 6 Discussion on Page 22, showing as: “To elucidate the synergistic interaction between HER and DIC, we compiled panel data from China's 31 provinces from 2011 to 2020. We employed the CCD model to measure their synergistic progress and utilized the Dagum Gini coefficient, kernel density estimation, traditional Markov chain, and spatial Markov chain methods to analyze regional disparities, temporal changes, and predictive trends of this synergy. We observed that the synergy between HER and DIC is moderate coordination, with good coordination provinces predominantly in the east, whereas the majority in the west are at a primary coordination. Notably, the eastern region exhibits the highest average level of synergy development. In contrast, the northeastern region, while on par with the national average, has been declining annually. Both the central and western regions lag behind the national average, yet they have each shown an upward trajectory since 2014. The observed regional disparities correlate with the uneven distribution of higher education resources in China [13], as well as the variability in the digital economy's development among different regions [15, 16]. The Dagum Gini coefficient demonstrates significant regional disparities in synergistic development, with inter-regional differences accounting for 67.07% of these disparities. The East-West divide exhibits the most pronounced differences, while the Central-Northeast divide is the least significant. Yet, the overall Gini coefficient exhibits a declining trend, suggesting that the disparities in regional synergistic development are diminishing. The results of this study are consistent with those reported by Shi [50]. The kernel density estimation confirms a reduction in the CCD imbalance. Specifically, the eastern region's CCD is moving leftward, with a reduction in absolute differences and a single peak, indicating no polarization. Synergistic development in the central region is increasing, with diminishing absolute differences and no polarization evident. The western region's trend fluctuates, initially decreasing before rising, consistently accompanied by a side peak and a gradient effect, indicating a polarization tendency. The north-eastern region's development is declining, yet absolute differences are reducing, with no evidence of polarization. Markov chain predictions indicate that synergistic development primarily transitions between adjacent types, which complicates achieving a significant leap in development. Middle and high-level neighborhoods enhance local synergistic development, while lower-level neighborhoods may impede progress and potentially reverse gains.”

4. The strengths and limitations of the study should be specially addressed, e.g., using sub-headings to address.

Response: Thank you for the suggestion. We have added the contributions and limitations required as explained above. Please read the Section 8 Contributions and limitations on Page 24, showing as: “This study offers insights into addressing disparities in digital higher education across regions. Distinct from other research on higher education and digitization, this study contributes to the digital education literature by: elucidating the integration mechanism between HER and DIC, assessing the synergistic development through the CCD model, and examining regional disparities, spatial distribution, and forecasting trends in this synergistic development, thereby establishing a theoretical foundation for subsequent studies and offering empirical evidence to address the imbalance in China's higher education sector. Nevertheless, the study has certain limitations: Firstly, as a provincial study, it suggests that future research could collect urban data for a more granular analysis of higher education's digital transformation. Secondly, the study has not yet investigated the driving factors behind this synergistic development; future research should address this omission.”

5. The conclusions and implications should be shortened and made more concise, some words and explanations should be relocated to discussion part.

Response: Thank you for the suggestion. We have modified the conclusions and implications required as explained above. Please read the Section 7.1 Conclusions on Page 23, showing as: “The integration of HER and DIC is vital to China's educational reform, and addressing regional disparities in their combined advancement is key to improving higher education quality. The key findings of this study are as follows: (1) The coordinated development of HER and DIC exhibits a fluctuating trend, initially declining then rising, with the eastern region demonstrating notably higher synergy compared to the other three, highlighting pronounced regional disparities. (2) The synergistic development of HER and DIC reveals regional disparities, with the eastern region having the greatest intra-regional difference and the most significant inter-regional difference occurring between East-West, primarily attributed to inter-regional differences. (3) While regional disparities in the synergistic development are diminishing, a weak trend toward multi-polarity is emerging, with polarization being particularly evident in the western region. (4) The synergistic development exhibits four distinct patterns of convergence, with the most pronounced convergence levels being of greater significance, and spatial factors exerting a considerable influence on regional synergy.”

6. The manuscript requires a language revision.

Response: Thank you for your advice, we have revised the whole manuscript carefully to avoid language errors. In addition, we have consulted a professional language-editing service and asked several colleagues who are native English speakers to check the English. We believe that the language is now acceptable for the review process.

Reviewer #2: The study titled “Synergy of Higher Education Resources and Digital Infrastructure Construction in China: Regional Differences, Dynamic Evolution and Trend Forecasting.” However, it needs a few minor revisions to improve its quality. My details comments are as below.

The abstract should present the research objectives, methods, and concise results of the study. Revise these concerns.

Response: Thank you for the suggestion. We have modified the abstract required as explained above. Please read the Section Abstract on Page 1, showing as: “The deep integration of higher education with digital technology represents an inevitable trend, and evaluating the interplay between higher education resources (HER) and digital infrastructure construction (DIC) holds significant value for advancing the development of digital higher education and mitigating regional disparities in China. This study establishes two comprehensive evaluation frameworks for HER and DIC. Panel data from 31 provinces, spanning the period from 2011 to 2020, are utilized for analysis. The coupling coordination degree (CCD) model is employed in this work to evaluate the synergy between HER and DIC in China. Furthermore, we analyze the regional differences, spatial distribution, and trend evolution of this synergy. The study results revealed that there is an initial decrease followed by an increase in the synergy between HER and DIC, and the overall CCD is at a moderate coordination, with the mean CCD of the eastern region being significantly higher than that of the other three regions, and the inter-regional difference is the main source of regional disparity in this synergy. The current state of synergistic development reveals a slight inclination towards multi-polarization, although the disparity in regional development was decreasing. Additionally, there is an observed convergence in the coordinated development of HER and DIC, with spatial factors playing a significant role. These findings offer empirical support for efforts to enhance the integration of HER and DIC, reduce regional disparities in higher education, and foster sustainable development in China's higher education sector.”

The introduction section is nicely developed. Mention the objectives in this section.

Response: Thank you for the compliment.

Rewrite the following paragraph in the introduction section for more clearance” Since the beginning of the century, an increasing number of universities have been developing "digital education" with the help of ICT [19]. A richer exploration of the link between higher education and digitization has been undertaken by scholars, including the following aspects: Firstly, the need for digital technologies in higher education. A multitude of academic investigations has consistently substantiated the fact that the COVID-19 pandemic has necessitated higher education establishments to adopt online learning platforms as a means to uphold their educational frameworks. This situation has underscored the critical importance of integrating digital technologies into the global education system [20-22]. Furthermore, to maximize student engagement in the classroom, Penprase (2018) contended that incorporating projectors, computers, and other advanced technological devices in higher education is likely to captivate students' interest, thereby promoting their active participation in classroom learning and offering them an immersive learning experience [23]. Additionally, Ozdamli and Cavus (2021) emphasized the significance of feedback loops in digital classrooms, as they enable students to receive timely feedback from their instructors in real time [24].

Response: Thank you for the advice. We have modified the part of text expression according to reviewer’s comment. The precedent version of this text expression has been replaced, becoming “An increasing number of universities have been developing ‘digital education’ with the help of ICT [17]. A richer exploration of the link between higher education and digitization has been undertaken by scholars, including the following aspects: Firstly, the need for digital technologies in higher education. A multitude of academic investigations have confirmed that the COVID-19 pandemic has compelled universities to embrace online platforms to maintain their educational systems, highlighting the need for digital integration in global education [18-20]. Penprase [21] argued that using advanced tech like projectors and computers in higher education can enhance student engagement and foster active participation by providing an immersive learning experience. Ozdamli and Cavus [22] highlighted the importance of feedback loops in digital classrooms, noting that they allow for real-time instructor feedback, which is crucial for student learning.”. Please read the first two paragraphs of the Literature review, on Page 2-3.

Add the following latest literature on the topic

Response: Thanks for the reviewer’s kind advice. According to the reviewer’s suggestion, we have added the two latest literature. Please read the second sentence of the second paragraph of the Introduction, on Page 2, showing as: “Evidence suggests that combining educational resources with more advanced technology will promote regional productivity growth [11]”. Also, please read the first sentence of the third paragraph of the Introduction, on Page 2, showing as: “Previous studies have found that there is a relatively obvious imbalance higher education development in China [12-14]”.

Check the table numbers.

Response: Thanks for the reviewer’s kind reminder. We have checked the table numbers again.

Page number is missing please add

Response: Thank you for the suggestion. We have added the page numbers.

Results should be backed by literature. Add more discussion and add citations to back your results.

Response: Thanks for the reviewer’s kind advice. We have added the discussion required as explained above. 

---

## [Decision Letter · Decision Letter 1]

15 May 2024

Synergy of higher education resources and digital infrastructure construction in China: regional differences, dynamic evolution and trend forecasting

PONE-D-24-03628R1

Dear Dr. Zhang,

We’re pleased to inform you that your manuscript has been judged scientifically suitable for publication and will be formally accepted for publication once it meets all outstanding technical requirements.

Kind regards,

Muhammad Farooq Umer, PhD Epidemiology and Health Statistics

Academic Editor

PLOS ONE

Additional Editor Comments (optional):

Reviewers' comments:

Reviewer's Responses to Questions

**Comments to the Author**

1. If the authors have adequately addressed your comments raised in a previous round of review and you feel that this manuscript is now acceptable for publication, you may indicate that here to bypass the “Comments to the Author” section, enter your conflict of interest statement in the “Confidential to Editor” section, and submit your "Accept" recommendation.

Reviewer #1: All comments have been addressed

Reviewer #2: All comments have been addressed

2. Is the manuscript technically sound, and do the data support the conclusions?

Reviewer #1: Yes

Reviewer #2: Yes

3. Has the statistical analysis been performed appropriately and rigorously? 

Reviewer #1: Yes

Reviewer #2: Yes

4. Have the authors made all data underlying the findings in their manuscript fully available?

Reviewer #1: No

Reviewer #2: Yes

5. Is the manuscript presented in an intelligible fashion and written in standard English?

Reviewer #1: Yes

Reviewer #2: Yes

6. Review Comments to the Author

Reviewer #1: The authors have revised the manuscript according to the reviewer's comments and the manuscript can be accepted after proofreading.

Reviewer #2: No more revision required. All the suggestions addressed, therefore i am recommending acceptance of manuscript.

7. PLOS authors have the option to publish the peer review history of their article (what does this mean?). If published, this will include your full peer review and any attached files.

Reviewer #1: No

Reviewer #2: No

---

## [Editor Report · Acceptance letter]

22 May 2024

PONE-D-24-03628R1 

PLOS ONE

Dear Dr. Zhang, 

I'm pleased to inform you that your manuscript has been deemed suitable for publication in PLOS ONE. Congratulations! Your manuscript is now being handed over to our production team.

Kind regards, 

on behalf of

Dr. Muhammad Farooq Umer 

Academic Editor

PLOS ONE